# Statistical analysis of observations of polar stratospheric clouds with a lidar in Kiruna, northern Sweden

Peter Voelger[1] and Peter Dalin[1]

[1]Swedish Institute of Space Physics (IRF), Kiruna, Sweden

**Correspondence:** Peter Voelger (peter.voelger@irf.se)

**Abstract.** In the present paper, we analyze 11 years of lidar measurements to derive general characteristics of Polar Stratospheric Clouds (PSCs) and to examine how mountain lee waves influence PSC properties. Measurements of PSCs were made with a backscatter lidar located in Kiruna, northern Sweden, in the lee of the Scandinavian mountain range. The statistical analysis demonstrates that nearly half of all observed PSCs consisted of nitric acid trihydrate (NAT) particles while ice clouds were only a small fraction, supercooled ternary solution (STS) and a mixture of different components making up the rest. Most PSCs were observed around 22 km altitude. Mountain lee waves provide a distinct influence on PSC chemical composition and cloud height distribution. Ice PSCs were about 5 times as frequent and NAT clouds were about half as frequent under wave conditions. PSCs were on average at 2 km higher altitudes when under the influence of mountain lee waves.

## 1 Introduction

Polar stratospheric clouds (PSCs) are a well-known feature of the polar wintertime stratosphere. Their influence on chemical reactions related to ozone depletion in the lower stratosphere was first postulated shortly after the discovery of the Antarctic ozone hole and has since then been subject of many studies (for reviews see e.g. Solomon (1999), Lowe and MacKenzie (2008), or Tritscher et al. (2021)). In short, the formation of PSC particles that contain nitric acid has been identified to accelerate ozone depletion by (a) enabling chemical reactions on the surface of cloud particles that form chlorine radicals, (b) removing gas-phase $HNO_3$ by the formation of cloud particles and (c) permanently depleting the stratosphere of $HNO_3$ through sedimentation of those particles. The impact a PSC has on ozone concentration depends on the composition of the cloud (Kirner et al., 2015; Tritscher et al., 2021). PSCs typically consist of nitric acid trihydrate (NAT, a compound consisting of nitric acid and water at a molar ratio of 1:3), supercooled ternary solution (STS, a mixture of nitric acid, sulfuric acid and water), or water ice. In the context of lidar measurements such clouds are also referred to as type Ia, Ib and II, respectively (Browell et al., 1990), as the backscatter signals from these different PSC types have very distinct characteristics. Additional subtypes have been proposed to explain observations that don't fit any of the above three types. These subtypes include modified components or mixtures of the basic cloud types (see e.g. Tabazadeh and Toon (1996), Stein et al. (1999), Massoli et al. (2006)). Based on laboratory studies additional compounds have been predicted to exist in PSCs but could, so far, not be identified in natural stratospheric clouds beyond any doubt (Tritscher et al. (2021) and references therein).

The significant contribution of PSCs to ozone depletion is one of the major reasons for investigating and monitoring these clouds. Satellites, due to the near global coverage of their observations, offer a way to study the worldwide distribution of PSCs. An early attempt to prepare a multi-year statistic based on satellite data was undertaken by Poole and Pitts (1994) who statistically analysed PSC occurence by means of limb measurements with the photometer SAM II on board of Nimbus 7. Datasets from other, more advanced instruments have become available in recent decades (Spang et al., 2018; Pitts et al.,

2018). Prominent examples are MIPAS (Michelson Interferometer for Passive Atmospheric Sounding) on board of Envisat (2002–2012) and CALIOP (Cloud-Aerosol Lidar with Orthogonal Polarization) on the CALIPSO satellite (since 2006). On the basis of MIPAS measurements it was possible to infer the chemical composition of PSCs. The MIPAS dataset has been utilised for long-term studies of the composition of PSCs (Spang et al., 2018). CALIOP collects backscatter signals from the atmosphere that give information about PSCs with better spatial resolution than what is possible with passive remote

sensing techniques, as e.g. MIPAS. Over the period of operation the lidar data has been used to generate multiple global PSC climatologies with the latest version covering 11 years of observations until 2017 (Pitts et al., 2018).

While satellite-borne instruments provide observations with nearly global coverage, ground-based lidar measurements are better suited to give insight in how the specific conditions at the location of the instrument affect PSCs. The time and vertical resolution of ground-based lidar observations are better than what can be achieved by spaceborne instruments, thus enabling

detailed studies of local pecularities. Several ground-based lidars have been or were operating over sufficiently long periods to allow for the statistical analysis of PSC characteristics at the respective locations. Santacesaria et al. (2001) and Tencé et al. (2022) compiled such data, although for different periods, for the lidar at the Dumont d'Urville research station in Antarctica. Adriani et al. (2004) published a similar survey for the McMurdo station, also in Antarctica. Studies for data from the Arctic region were published by Massoli et al. (2006) for Ny-Ålesund, Spitsbergen, and by Blum et al. (2005) and Achtert and Tesche

(2014), both the latter examined more than a decade of measurements with a lidar at Esrange in northern Sweden.

Atmospheric waves are known to create temperature pertubation due to adiabatic cooling/warming when a wave forces an air parcel to move vertically. In the stratosphere such changes of the temperature can affect PSC formation and composition. As most prominent sources for waves have been identified topography, convection and adjustment of unbalanced flow near jet streams and frontal systems (Hoffmann et al., 2017). Depending on the horizontal wavelength atmospheric waves are classified

as planetary, synoptic-scale, or gravity waves. While the boundaries between these types are not exactly defined, it is generally assumed that the former comprises horizontal wavelengths of a few 1000 km or more, the latter of less than several 100 km, and the synoptical scale covering the range in between. Of the gravity wave type, inertia gravity waves (horizontal wavelengths $> 10$ km) are significant, since they are not trapped at low altitudes but can propagate up to the stratosphere and even the mesosphere (Fritts and Alexander, 2003). Utilizing analysis data that was produced by the European Centre for Medium-Range Forecast

(ECMWF) Teitelbaum and Sadourny (1998) showed that the distribution of temperatures that allow the formation of PSCs in the Arctic region is, to a large extent, correlated with the upwelling of isentropic surfaces as a result of planetary waves. Kohma and Sato (2011) used, among other data sets, CALIPSO observations to examine the extent to which waves of different scales influence the formation of PSCs. They deduced that stationary planetary waves are the dominant factor in the formation of PSCs in the Arctic while synoptic-scale waves only have small influence. They further concluded that inertia gravity waves

have negligible impact on Arctic PSC formation. Their reasoning, however, is based on data from only one winter season which, given the large interannual variability of Arctic winter conditions, is a too short period to draw definite conclusions. On the other hand, Alexander et al. (2013), using CALIPSO and other satellite data spanning four winter seasons, determined that mountain lee waves, a subtype of inertia gravity waves, significantly influenced PSC composition and were responsible for nearly 1/3 of all PSC occurences in the Arctic. The differences between the conclusions by Kohma and Sato (2011) and those by Alexander et al. (2013) indicate that, due to large interannual variability, statistically sound conclusions have to be based on a data base that spans at least several years, preferably a decade or more.

Mountain lee waves are typically generated when air is vertically displaced while wind is blowing across mountains. Under favourable conditions such waves can propagate from the surface up to the mesosphere. The stratospheric temperature pertubations that are associated with them can be as much as 10 K or more (Dörnbrack et al., 1997). Hence, they can trigger both the formation of a PSC (Voelger and Dalin, 2021) and the change of the PSC composition (Carslaw et al., 1999). Although mountain lee waves are regionally confined phenomena they can set off the formation of meso-scale PSCs (Carslaw et al., 1998b; Eckermann et al., 2009). Chemical reactions on the surfaces of PSC particles can then cause large-scale ozone depletion. Pitts et al. (2011) showed that CALIPSO data can be used to detect the impact of mountain lee waves on PSCs and Alexander et al. (2013) utilised this capability for estimating the role of these waves in PSC formation. However, Alexander et al. (2013) pointed out that CALIPSO can miss the detection of mountain lee waves, as only a small, random portion of a wave might be sampled which not necessarily shows the backscatter characteristics which were defined for identifying waves in CALIPSO observations. Moreover, the low sampling frequency results in that many cases with waves being present are missed. For fixed locations observations with a ground-based lidar can therefore give more detailed data.

Kiruna, the location of our lidar, is situated on eastern slopes of the northern part of the Scandinavian mountain range, a region where mountain lee waves frequently occur (Rao et al., 2008). Several field campaigns have been conducted in northern Scandinavia in the past decades which investigated cases when mountain lee waves influenced composition and geometry of PSCs (see e.g. Tsias et al., 1999; Dörnbrack et al., 2002; Lowe et al., 2006; Dörnbrack et al., 2012; Molleker et al., 2014). The aim of our study is to present a systematic investigation of the impact mountain lee waves have on PSC characteristics at a location where such waves are frequently present. We will use data from 11 years of measurements with our lidar (in the following also referred to as *IRF lidar*) both to derive general characteristics of the observed PSCs and to examine how mountain lee waves influence PSC properties. We first shortly describe the lidar and the data that has been used in this study. We then derive some general characteristics of PSCs over our location and compare them with results from similar studies. Thereafter we discuss, with help of a statistical analysis of the data, how mountain lee waves modified PSCs above Kiruna.

## 2  Data and Method

Lidar observations were performed with a backscatter lidar that is located on the premises of the Swedish Institute of Space Physics (IRF) in Kiruna at $67.84°$N, $20.41°$E. The lidar operates at 532 nm wavelength and has two detection channels to distinguish backscatter signals with two orthogonal planes of polarisation. Light with the same plane of polarisation as the laser

will in the following be denoted as parallel while perpendicular will refer to light with a polarisation plane perpendicular to that of the laser. Height and time resolution are 30 m and 133 s, respectively. The altitude range for observations is from 5 to 50 km (see Voelger and Nikulin (2005) for a more detailed description of the lidar characteristics). Measurements were performed during winters from 2007/08 to 2017/18. Specifics on the distribution of observations during this period will be discussed in the next section. Measurements were carried out whenever conditions were favourable for the presence of PSCs, i.e. predicted temperatures in the stratosphere were approximately around PSC existence temperatures or lower, and weather conditions allowed for lidar measurements of the stratosphere. Temperature predictions were provided by the Danish Meteorological Institute, based on forecasts by the ECMWF. Lidar observations were restricted to nighttime and twilight for better signal quality. For the subsequent analysis we integrated our measurement data over time to derive hourly averages. Additionally, a 5-point moving average over altitude was applied.

The signal that the lidar detects is proportional to the backscattering that is caused by molecules, aerosol particles, and cloud particles in the atmosphere. When considering the stratosphere the contribution of particles to the backscattering at 532 nm from a cloud-free atmosphere (i.e. outside PSCs) can be assumed to be a few percent only (Thomason et al., 2007). In the context of PSC observations the backscattering from both molecules and aerosol particles combined represent the background signal.

Parameters that are commonly used to describe PSCs in lidar measurements are the backscatter ratio of the parallel channel, $R_\parallel$, and the depolarisation ratio $\delta$. We define $R_\parallel$ as the ratio of total backscatter coefficient to that of the atmospheric background, both for the parallel channel:

$$R_\parallel = \frac{\beta_{\parallel,PSC} + \beta_{\parallel,BG}}{\beta_{\parallel,BG}} \tag{1}$$

with $\beta$ being the backscatter coefficient and indices $PSC$ and $BG$ denoting cloud and background, respectively. As background we consider contributions from both molecules and stratospheric aerosol particles. $R$ larger than 1 implies that a cloud is present. The backscatter coefficient of the background can be assumed to be proportional to the interpolated signals from below and above the PSC. Similarly, the cumulative backscatter coefficients of background and PSCs is proportional to the total backscatter signal. Hence, $R_\parallel$ can be calculated as

$$R_\parallel = \frac{I_{\parallel,tot}}{I_{\parallel,BG}} \tag{2}$$

where $I_{\parallel,tot}$ denotes the total backscatter signal and $I_{\parallel,BG}$ the interpolated signal without PSC, both for the parallel channel. The backscatter ratio for the perpendicular channel, $R_\perp$, can be estimated in an analogous way. From the backscatter ratios for both channels the depolarisation ratio can be determined:

$$\delta = \frac{R_\perp}{R_\parallel} \delta_{mol} \tag{3}$$

where $\delta_{mol}$ is the depolarisation ratio of molecular backscattering. We assume that stratospheric background aerosol particles are predominantly sulfuric acid droplets which are spherical particles and, therefore, do not change the polarisation of light that is scattered on them at 180° (i.e. backscattering). For our lidar we estimate $\delta_{mol}$ as 0.02. The backscatter signals in both

receiver channels can be skewed by (a) cross talk between both channels and (b) polarising effects of the receiver optics. We examined the error from both sources by using white light with a controlled state of polarisation (method described by Mattis et al. (2009)). Our tests showed that the errors for our lidar are negligible.

    For evaluation of the atmospheric conditions that were prevalent during lidar measurements we used horizontal wind fields from the ERA5 dataset (Hersbach et al., 2017) that can be obtained from ECMWF (Hersbach et al., 2018; Hersbach et al.,
2020). Such data is available for every full hour with a horizontal resolution of 31 km. The data comes at 137 vertical levels between surface and 1 Pa pressure level since mid 2013 and at 91 levels before.

## 3   Data analysis

### 3.1   General PSC characteristics

The lidar was operated during 11 winter seasons, starting in late 2007 and ending in early 2018. During this period mea-
surements of PSCs were performed in 102 nights. In total, 738 hours of observations were accumulated. The distribution of measurement times per year is shown in Figure 1. The number of nights with observations per season was varying between 2 nights during winter 2014/15 and 19 nights (2015/16). To a large extent, the year-to-year variation is a consequence of interannual variations of location and strength of the polar vortex and of the temperatures inside of it. The influence of vortex conditions becomes apparent when comparing the two winters mentioned above, 2014/15 and 2015/16. The stratosphere
during winter 2014/15 was relatively warm (Wohltmann et al., 2020), resulting in that both the period when conditions were favourable for PSC formation was relatively short and that the PSC areal coverage was small. As a consequence, at our location PSCs were present only during a few nights. On the other hand, during winter 2015/16 the polar vortex was unusally cold and stable (Manney and Lawrence, 2016; Matthias et al., 2016; Khosrawi et al., 2017). This resulted in a very large PSC areal coverage over a long period and, therefore, more opportunities for ground-based lidar observations of PSCs.

A second factor that limits the number of observations are tropospheric clouds, i.e. weather conditions. If those clouds are optically too thick, they prohibit measurements of stratospheric features from the ground. Weather conditions are highly variable, therefore number and length of periods of sufficiently good conditions for stratospheric observations changes from year to year. The influence of tropospheric clouds becomes obvious when comparing the number of days with measurements during winters 2012/13 and 2013/14. The number of observation events and the total number of measurement hours were the
same in both winters. Yet, the stratospheric conditions were different. The polar vortex during winter 2012/13 was relatively weak and broke up already in early January while during the following winter the vortex was stronger and lasted until late January (Lawrence and Manney, 2018). Hence, conditions that were favourable for PSCs occured more often during 2013/14. That these differences are not reflected in the number of our measurements has to be explained by differences in the prevailing tropospheric conditions. While cloudiness of the troposphere does affect annual measurement statistics to various degree, we
expect the influence to average out over the period that is included in this study.

    The earliest measurement during a winter season was on November, 30th (2017), the latest on February 18th (2017). About 60% of all measurements of PSCs were during January, 27% and 12% during December and February, respectively. As PSCs,

| Month | Days with $T < T_{NAT}$ | lidar observations |
|-------|------------------------|--------------------|
| Nov | 9 (2.4%) | 1 (1.0%) |
| Dec | 116 (31.2%) | 28 (27.5%) |
| Jan | 167 (44.9%) | 61 (59.8%) |
| Feb | 61 (16.4%) | 12 (11.8%) |
| Mar | 17 (4.6%) | 0 (0%) |
| Total | 370 (100%) | 102 (100%) |

**Table 1.** Number of nights when temperatures over Kiruna (derived from ECMWF analysis data) were sufficiently low to allow for PSCs to exist (left column) and number of nights when lidar measurements were performed (right column). The numbers are for the whole period Nov 2007– Mar 2018.

when observing from the ground, sometimes are obscured by tropospheric clouds the exact number of nights with PSCs present over our location is unknown. However, stratospheric temperatures can be used as an indicator for periods that were sufficiently

cold for PSCs to exist. Based on the thermodynamic equilibrium equations for NAT that were formulated by Hanson and Mauersberger (1988) and on the daily analysis of atmospheric conditions prepared by the ECMWF, the Danish Meteorological Institute composes maps of the Arctic region for certain altitudes that indicate areas where PSCs can exist (i.e. $T < T_{NAT}$)[1]. Table 1 shows the monthly numbers of nights with $T < T_{NAT}$ for the period of our observations and the monthly number of nights with actual lidar measurements. Based on temperature, observations during January are slightly overrepresented in our

data while measurements during November and December are relatively rare. A likely reason is that the troposphere during those months is less stable and tropospheric clouds are more common. Hence, opportunities for lidar measurements are less frequent. Similarly, during later winter (months February and March) we performed relatively few measurements in relation to the number of nights when temperatures were below the calculated existence temperatures for NAT. Here, the reason most probably is that for calculating $T_{NAT}$ a constant concentration of the NAT precursor $HNO_3$ during the whole winter is assumed,

and, hence, $T_{NAT}$ is constant as well, while actual concentrations of $HNO_3$ are decreasing over the winter due to denitrification of the stratosphere as a result of PSC formation. Lower concentrations of $HNO_3$ then require lower temperatures for particles to form.

For all available data of lidar measurements hourly averages were calculated and included in the following analysis. This is different from the statistical analysis that Adriani et al. (2004) presented in their study of PSCs at the McMurdo station.

Adriani et al. (2004) utilised only one profile per Julian day in order to avoid a possible bias as observation periods may vary from day to day. This approach presumes that PSCs show only minor variations during a day so that one hourly profile can be considered as representative for the whole 24 hour period. Since mountain lee waves are frequently present at our location,

---

[1]http://psc.dmi.dk

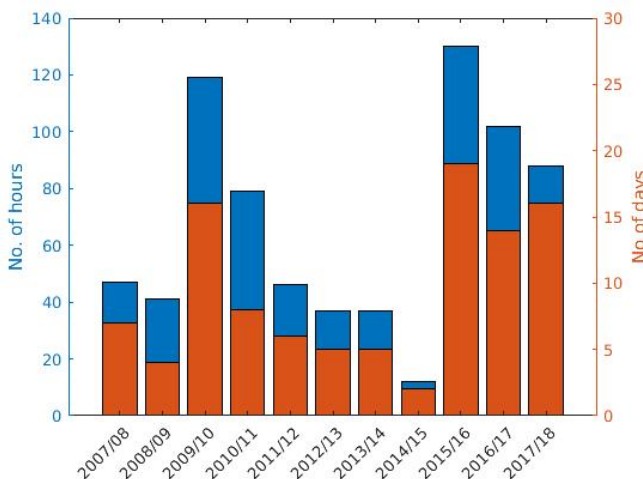

**Figure 1.** Annual statistic of IRF lidar measurements. Blue marks the hours of measurements per year, red bars are the number of nights with PSC observations.

PSC characteristics can change significantly within a short time. Using only one profile per Julian day would not account for the variations that can occur during that period, or, in our case, over the course of a night.

The characteristics of the backscatter signal from a PSC depend on the type of particles in the cloud. Combining signal strength, or as a derived parameter, the backscatter ratio, with the depolarisation of the backscatter makes it possible to classify PSCs according to their composition. However, parameter ranges depend in part on the lidar system and the wavelength(s) it utilises. Several sets of criteria to classify observed clouds have been suggested in the past (see e.g. Browell et al., 1990; Biele et al., 2001; Pitts et al., 2018). A thorough comparison of these and a few other classification schemes was conducted by

Achtert and Tesche (2014). They pointed out that PSC classifications were generally developed with the purpose of interpreting a certain set of data and that, hence, thresholds for PSC types were set with respect to instrumental constraints, measurement conditions, and which PSC-related parameters were available. Achtert and Tesche (2014) concluded that in order to better be able to distinguish PSC types other than ice, the classification scheme should be based on the backscatter ratios for both planes of polarisation instead of the total ratio. They further argue that of the PSC classifications that only use lidar observation data

the scheme proposed by Blum et al. (2005) has the best performance.

    Our lidar is similar in design to the one used by Blum et al. (2005) for their long-term study. Therefore our classification is based on the same parameters as theirs, i.e. $R_\parallel$ and $\delta$. However, we slightly modified the thresholds for the depolarisation ratio. Since our lidar utilises other, wider bandwidth filters to suppress atmospheric noise, we have to account for a larger background depolarisation. For the backscattered signal to be interpreted as being influenced by the presence of PSCs the backscatter ratios

in either detection channel should be clearly distinguishable from the background. For our lidar data we set lower limits to 1.05 and 1.1 for the parallel and the perpendicular channel, respectively. These numbers mean that we potentially miss a few very thin PSCs. However, in the vast number of cases cloud boundaries were not sensitive to the lower limits for $R$. The values used

| PSC type | | Backscatter ratio $R_{\parallel}$ | Depolarisation ratio $\delta$ |
|---|---|---|---|
| NAT (type Ia) | | $1.05 < R_{\parallel} < 2$ | $\delta > 10\%$ |
| STS (type Ib) | | $1.1 < R_{\parallel} < 5$ | $\delta < 2\%$ |
| Ice (type II) | either | $2 < R_{\parallel} < 7$ | $\delta > 3\%$ |
| | or | $R_{\parallel} > 7$ | any |

**Table 2.** Classification criteria for PSCs observed with the IRF lidar.

in our classification scheme are summarised in Table 2. Combinations of $R_{\parallel}$ and $\delta$ that don't match any particular PSC type are assumed to consist of a mixture of different types of particles. It should be noted that setting of fixed boundaries for individual types means a simplification of actual atmospheric conditions. This can lead to misclassification of measurement data. Pitts et al. (2013) and Pitts et al. (2018) discussed this issue for data inverted from CALIOP observations where the problem is mainly due to signal noise. In our case, temporal and spatial averaging will mean that fine PSC structures, e.g. filament layers, can get blurred. Furthermore, a PSC that is subject to vertical motion due to mountain lee waves will experience temperature changes of several Kelvin. During that motion the PSC is out of its thermodynamic equilibium which in turn results in fast changes of the composition. Mountain lee waves are seldom stationary, a consequence of the complex topography and wind changes that generate them. Therefore, averaging over time can cover different parts of the wave motion. Finally, Biele et al. (2001) and Pitts et al. (2009) pointed out that PSCs showing optical charateristics of a certain type can still contain some, minor amount of other compounds as well.

Figure 2 shows a 2-D histogram of the backscatter ratios for parallel and perpendicular polarisation for PSC increments of all our measurements. The boundaries of the various PSC types are based on those proposed by Blum et al. (2005) but adapted for the specifications of the IRF lidar. They are shown in Figure 2, as well. Two clusters with large numbers of observations become obvious, one with small $R_{\parallel}$ and large $R_{\perp}$ (i.e. large $\delta$) and one with small $R_{\perp}$ and moderately large $R_{\parallel}$ (i.e. small $\delta$). The first cluster represents observations with the characteristics of PSCs consisting of NAT particles while the latter is typical for PSCs with STS. Additionally, a third, smaller cluster of measurements exists with large $R$ in both channels, indicating ice PSCs. Moreover, the histogram shows that a considerable number of data points are found between those three clusters. This can be interpreted as that the observed PSCs in many instances consisted of combinations of compounds instead of being predominantly composed of NAT, STS, or ice. Therefore, the boundaries between PSC types are in reality somewhat blurry. Mixtures of STS with one of the other types result in moderate depolarisation ratios and will be summarised as an additional type, called 'Mix'. The distinction between NAT and ice PSCs is not obvious, as both cause significant depolarisation ratios. From the point of lidar measurements, the principal difference between both types is the valid range for $R_{\parallel}$. However, it is well established that ice particles can act as seeds for the nucleation of NAT (see e.g. Carslaw et al., 1998a; Hoyle et al., 2013). Therefore, hybrids with according parameters can be expected at a certain rate at and near the boundary between both types.

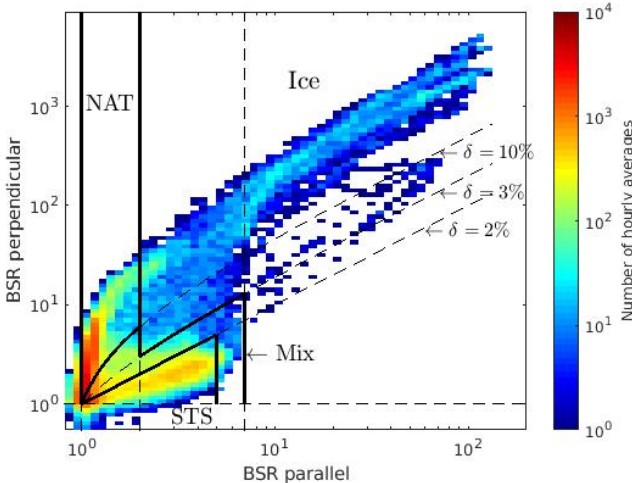

**Figure 2.** Frequency of backscatter ratios (BSR) of parallel versus perpendicular channel for all PSC observations with the IRF lidar. Parameter ranges for PSC types NAT, STS, and Ice are indicated by thick, solid lines They are based on the ranges that were proposed by Blum et al. (2005) but adapted for the specifications of the IRF lidar. Lines for some constant depolarisation ratio $\delta$ and some constant backscatter ratios are added for reference (dashed lines).

Displaying all our measurements as a piechart (Figure 3) shows that NAT was most common with almost half of all measurement points (46%). Ice amounted to 6% of the observations while STS and a mixture of particles occured in 26% and

21% of all cases, respectively. The large portion of NAT clouds can be attributed to the fact that NAT particles can exist at relatively warm temperatures (acc. to Hanson and Mauersberger (1988) at up to 7 K above the frost point of ice, $T_{ice}$). STS particles require lower stratospheric temperatures to exist, Carslaw et al. (1995) determined the equilibrium temperature to be appr. $T_{ice} + 4K$. Temperatures sufficiently low for ice PSCs to exist are rarely reached in the Arctic stratosphere, therefore such PSCs are not common. Compared to Pitts et al. (2018) our data shows less NAT PSCs and more mixed-type clouds. A

possible reason could be that Pitts et al. (2018) included data for the whole Arctic region while ours are from a single location that frequently is situated downwind of a major orographic obstacle, the Scandinavian Mountains. A climatology that Massoli et al. (2006) compiled for lidar observations at Ny-Ålesund, Spitsbergen found PSC types liquid (i.e. STS), NAT (including subtypes), and mixed to occur roughly equally frequently, with liquid being slightly more common. Since Ny-Ålesund is located further north, and, on average, closer to the center of the polar vortex than our location, colder stratospheric temperatures

can be expected there, leading to more frequent conditions that allow STS to exist. The results presented by Blum et al. (2005) and by Achtert and Tesche (2014) are based on lidar measurements at Esrange in Northern Sweden, a location situated roughly 40 km from our lidar. Hence, atmospheric conditions at both measurement sites are fairly similar. Nevertheless, PSC statistics differ to some extent. A reason for those differences could be that their measurements were being performed during campaigns, mostly between end of December and end of January when the Arctic polar vortex often is in its coldest phase. On the other

hand, our measurements were done throughout the whole PSC season whenever conditions allowed for lidar observations.

Therefore, our measurements were more often covering periods when the polar vortex was less well developed and, hence, the stratosphere was relatively warm.

The height distribution of our PSC observations is shown in Figure 4. For statistical purposes all cloud pixels were integrated into 1 km height intervals. In all cases when cirrus clouds were present during measurements they were spatially separated from the lowest PSC layer by at least a few km. Hence, an erroneous classification of a cirrus cloud as PSC can be excluded for our data. The lowest PSCs that were observed at our location were at altitudes between 14 and 15 km, the highest were between 33 and 34 km (The fraction of cloud pixels in the highest interval is too small to be distinguished from the y-axis of Figure 4.). Most frequent occurences were at altitudes of 21 to 23 km where approximately 1/4 of all PSCs were observed. The mean height was 21.48 km, standard deviation of the height distribution 3.04 km (see Table 3). Mean and median height were very similar, indicating that the height distribution was fairly symmetric around the mean height and close to the normal distribution. The height distribution of our data is consistent with what was reported by Pitts et al. (2018) who, based on 11 years of CALIPSO observations, showed that, for the Arctic region as the whole, the PSC aereal coverage is largest between 19 and 23 km.

Similarly, height distributions can be compiled for each PSC type separately (Figure 5). For all types of our classification, except for ice, the mean height of PSCs is between 21 and 22 km. The higher mean height for ice PSCs is a result of those clouds being primarily observed when gravity wave conditions prevailed. This will be discussed further down. The standard deviation $\sigma$, a measure for the width of the height distribution, is largest for NAT and Mixed PSCs. These two types also have large portions of cloud pixels at altitudes below 20 km (35% and 37% of observed PSC of respective type). Additionally, for both types mean and median of the respective height distributions are similar, meaning they are close to the normal distribution. On the other hand, for ice and for mixed PSCs the medians of the height distributions are smaller than the respective means, indicating that they are skewed towards larger heights.

The relative frequency of PSC types differed with altitude and is presented in Figure 6. NAT PSCs were prevalent up to 26 km while above mixed PSCs were more common. Ice PSCs were rarely observed below 20 km but the most common PSC type between 28 and 30 km. It should, however, be emphasised that only a small fraction of all observed PSCs were found at such high altitudes while the vast majority was detected at lower altitudes (Figure 4). This means that temperatures at these altitudes, compared to lower altitudes are far less frequently suffiently cold for any kind of PSC to occur. However, when ambient conditions do allow for PSCs to exist, then temperatures are often cold enough for Ice PSCs to form. This can be the consequence of the impact of mountain lee waves on PSCs. The effect of waves on PSC characteristics will be discussed in the following.

## 3.2 Wave influence on PSCs

The Scandinavian mountain range is known to be one of the major sources for mountain lee waves in the Arctic region (Hoffmann et al., 2017). This circumstance has triggered several case studies, addressing questions related to the generation of waves in northern Scandinavia and how they affect the atmosphere, in particular, the formation of PSCs (see e.g. Tsias et al.

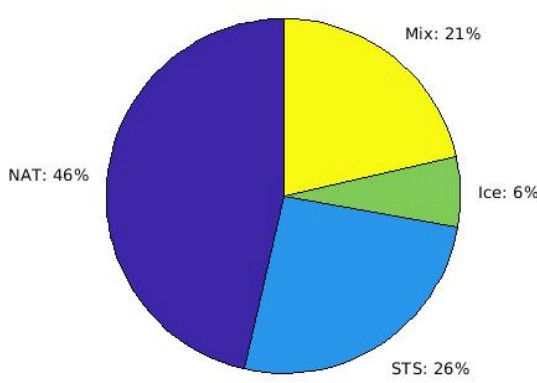

**Figure 3.** Relative distribution of PSC types for all measurements.

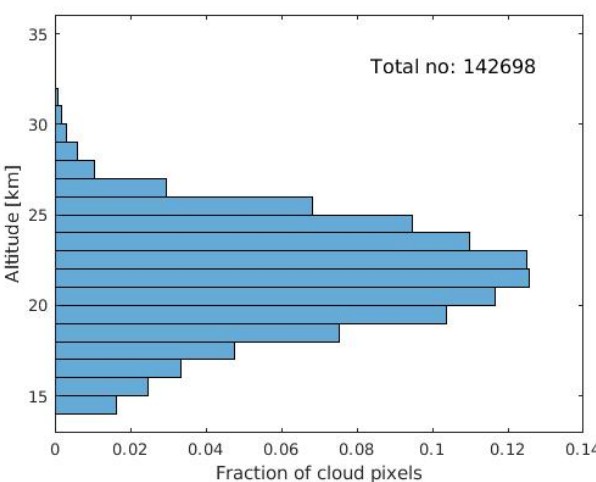

**Figure 4.** Height distribution of the relative fraction of detected PSCs. The total number of cloud pixels that are included is shown in the top right corner of the figure.

(1999) and Dörnbrack et al. (2002)). For such waves to be propagating from the lower troposphere up to the stratosphere certain
275 meteorological conditions need to be fulfilled. Dörnbrack et al. (2001) summarised them as follows:

1. The horizontal wind speed at 900 hPa has to be larger than a threshold value: $v_{hor}(900hPa) > v_{crit}$.

2. The direction of the horizontal wind at 900 hPa, $\alpha_{hor}(900hPa)$, must be approximately perpendicular to the mountain
   ridge: $\alpha_{mnt} - \Delta\alpha < \alpha_{hor}(900hPa) < \alpha_{mnt} + \Delta\alpha$.

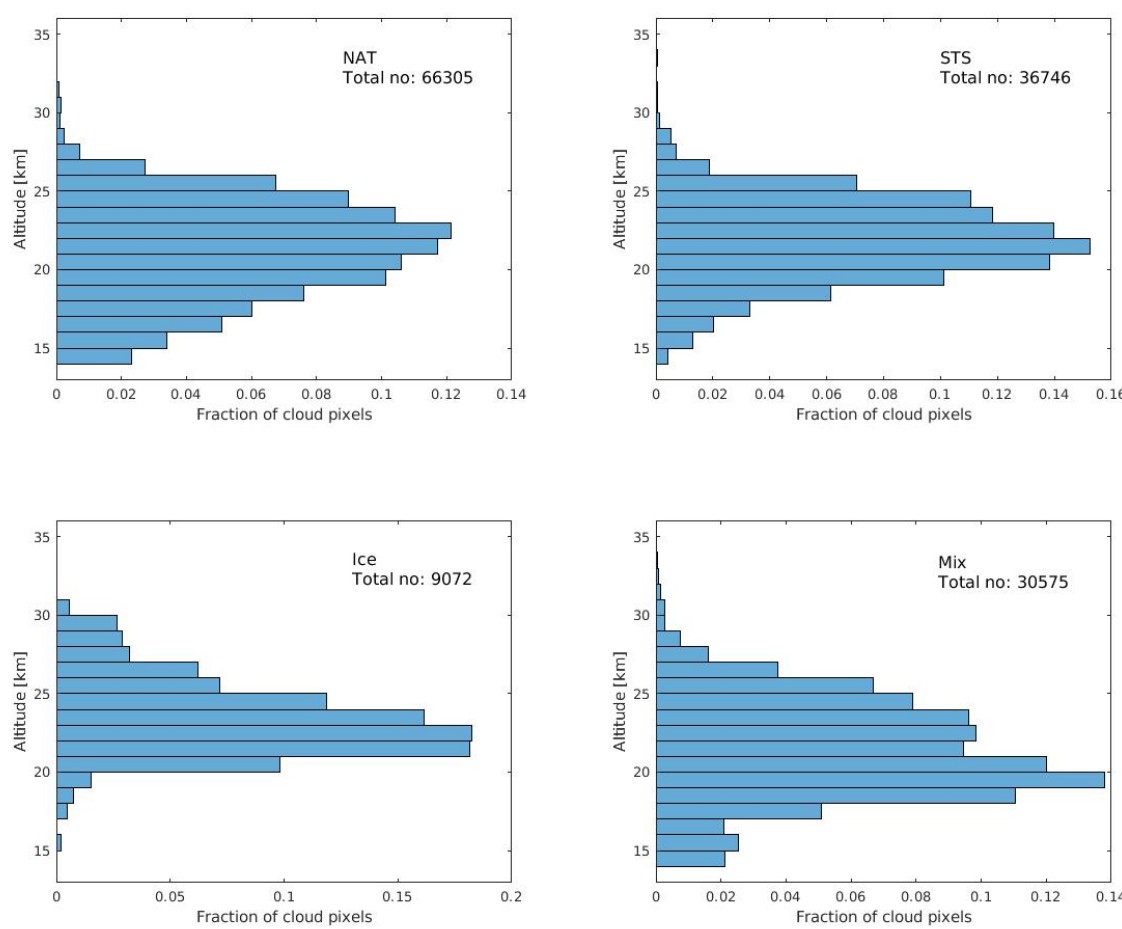

**Figure 5.** Same as Figure 4 but for the four types of our PSC classification scheme separately.

|  | all | NAT | STS | Ice | Mix |
|---|---|---|---|---|---|
| $\mu$ [km] | 21.48 | 21.13 | 21.79 | 23.43 | 21.30 |
| $Me$ [km] | 21.61 | 21.34 | 21.82 | 23.05 | 21.04 |
| $\sigma$ [km] | 3.04 | 3.14 | 2.61 | 2.43 | 3.20 |

**Table 3.** Mean height ($\mu$), median height ($Me$), and the standard deviation of the height distribution ($\sigma$), the latter being a measure for the width of height distribution, for our whole data set. Numbers are given both for all PSCs combined and for the individual types separately.

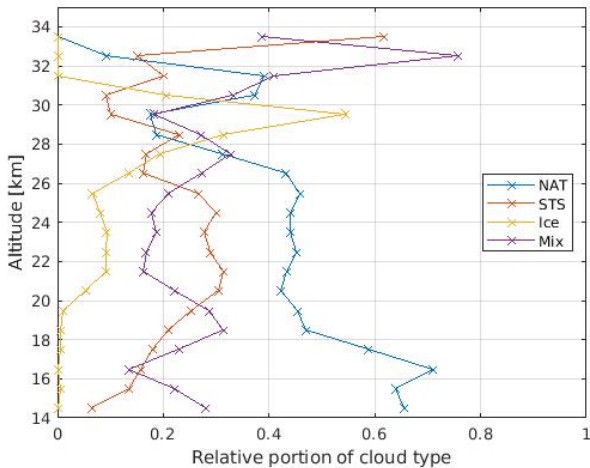

**Figure 6.** Relative portions of PSC types NAT, STS and Ice as function of altitude. 'Mix' refers to PSCs with optical characteristics that don't fit any of the above three types.

3. The wind direction at higher altitudes (pressure levels) doesn't differ much from that at 900 hPa: $\delta\alpha(p) = \alpha_{hor}(p) - \alpha_{hor}(900hPa) < \Delta\alpha$, with $p = 500, 300, 100, 50$ hPa.

In the case of the Scandinavian mountain range the orientation of the ridge in northern Scandinavia is from NNE to SSW at an angle of about $30°$ clockwise to the direct north. Hence, the wind direction perpendicular to the mountain range is $\alpha_{mnt} = 300°$. However, since the mountain range is not perfectly aligned even wind directions that are off the nominal normal by a certain angle $\Delta\alpha$ can still lead to lee waves. We set the thresholds to $\Delta\alpha = 45°$ and $v_{crit} = 10m/s$, the same value as suggested by Dörnbrack et al. (2001).

Wind data were extracted from the ERA5 dataset. Generally it is assumed that ERA5 data represents the observed atmosphere very well (Graham et al., 2019; Hersbach et al., 2018; Hersbach et al., 2018; Sivan et al., 2021). However, it should be noted that, for the stratosphere, the fit of ERA5 data to observations tend to be slightly worse than for the predecessor ERA-Interim (Hersbach et al., 2020). Comparisons for the Arctic troposphere on the other hand showed a better fit of ERA5 than for other global reanalyses (Graham et al., 2019). The horizontal resolution of ERA5 (appr. 30km) permits the proper representation of synoptic-scale winds but is too coarse to resolve small-scale disturbances. Yet, for verifying that atmospheric conditions allowed for mountain lee waves both to be generated and to propagate vertically, examining the synoptic wind field is sufficient.

A combination of all three criteria was applied to identify PSC measurements that were influenced by mountain lee waves. Figure 7 shows the year-to-year distribution of observation hours, divided into those with conditions that allowed mountain lee waves to propagate to the stratosphere and those that did not. Accumulated over the whole period, 230 hours of measurements were with mountain lee waves present, whereas 508 hours were without such waves. However, the interannual variability is large, both regarding the total numbers of days and hours with measurements and the portion of the measurements that

were influenced by waves. The latter varied between 0 and 70% which can be attributed to year-to-year variations of the wintertime polar troposphere. The result of these variations is that the chances for conditions that are favourable for gravity wave propagation differ largely over the years. Reasons for the variation of the total observation time were discussed further above.

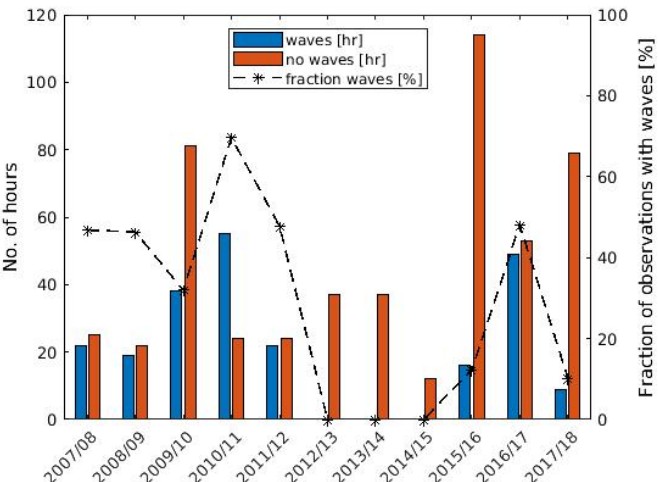

**Figure 7.** Annual statistic of hours of measurements with the IRF lidar. Blue bars mark times with conditions favourable for mountain lee waves based on criteria established by Dörnbrack et al. (2001), red bars are all other instances.

It has to be noted that inertia gravity waves can also be generated by sources other than topography, such as wind shears and wind jets which again can be caused by tropospheric pressure systems and the polar jet stream. Just as mountain lee waves, the temperature pertubations that are triggered by these waves can contribute to the modification or formation of PSCs (Hitchman et al., 2003; Shibata et al., 2003). However, the sources for these waves are not specific for our location but can also be found in most other parts of the polar regions, as well. Therefore, the impact of non-orographic waves is not a subject of this study.

The separation of wave and no-wave cases leads to two very different distributions of observed backscatter ratios, as shown in Figure 8. A significant portion of PSC observations during wave conditions were characterised by a large numbers of backscatter ratios for both states of polarisation, a manifestation of the presence of ice particles (Figure 8b). Such observations were absent for measurements that were not influenced by mountain lee waves (Figure 8a). Apparently, the presence of waves was a precondition for the formation of the vast majority of ice PSCs at our location in northern Scandinavia. On the other hand, observations of PSCs containing NAT were less frequent in wave conditions. This becomes even more obvious when visualising the portions of different PSC types as a pie chart for conditions with and without mountain lee waves separately. Apparently, the relative occurences of PSC types are notably different for wave and no-wave conditions (Figure 9). In the case of no waves present NAT made up the majority of all PSC measurements (53%), while ice was rarely observed (3%). This agrees well with findings by Pitts et al. (2018) for the whole set of CALIPSO measurements of the Arctic region for an 11-year period. On the other hand, when conditions for mountain lee waves were favourable, NAT, STS, and mixed clouds were

observed at approximately the same frequency (27–30% each). Ice clouds amounted to about 1/6 off all measurements. NAT PSCs were relatively less common, ice and mixed PSCs on the other hand relatively more frequent under wave conditions.

320    Another distinct effect of the influence of mountain lee waves is a change of the altitude of PSCs. Figure 10 shows the altitude distributions of observed clouds, separated for cases with influence of mountain lee waves (right panel) and without (left panel). Apparently, in the presence of mountain lee waves, the mean height of our PSC observations was moved to larger heights by appr. 2 km (see Table 4). Cloud bottoms and tops show a similar change, indicating that cloud layers as a whole shifted to larger heights. An explanation for this characteristic is that the lifting of air parcels due to the vertical motion in the
325   wave results in adiabatic cooling which leads to cloud formation around the wave crest. A similar process can be observed for tropospheric clouds when they are modulated by gravity waves.

In the same way as for the total data set before, height distributions can be compiled broken down by PSC type. This is shown in Figure 11 for measurements when no waves were present and in Figure 12 with mountain lee waves apparent. The corresponding mean and median heights and the widths of the height distributions are summarised in Table 4. For both NAT
330   and STS PSCs the mean heights during wave conditions were appr. 0.5 km larger than without waves present. In contrast, for ice and mixed PSCs the mean heights were appr. 2 km larger when waves were present. For all PSC types, except ice, the height distributions were wider in conditions without waves. Here the explanation is that with no wave influence a considerable number of PSCs were at heights below 20 km while such obervations were much less frequent when PSCs were influenced by mountain lee waves. A peculiar characteristic of the height distribution during wave conditions for mixed PSCs, and to a much
335   smaller extent also for NAT PSCs, is that it contains two maxima, one near 24 km and a second below 20 km. The minimum between both maxima is located at approximately the same height as the maximum of the distribution for ice PSCs, at 22–23 km. This suggests a connection between both features. The underlying reason is found in the different existence temperatures of the PSC types. Ice PSCs require the lowest temperatures while other types can exist in warmer air. Hence, ice PSCs are frequently found surrounded by other types of PSCs (see e.g. Dörnbrack et al., 2002; Shibata et al., 2003; Achtert and Tesche, 2014; Pitts et al., 2018).

| | all | | NAT | | STS | | Ice | | Mix | |
|---|---|---|---|---|---|---|---|---|---|---|
| | nw | wa | nw | wa | nw | wa | nw | wa | nw | wa |
| $\mu$ [km] | 20.93 | 22.82 | 22.66 | 23.12 | 21.64 | 22.15 | 22.09 | 23.95 | 20.57 | 22.52 |
| $Me$ [km] | 21.07 | 22.87 | 20.86 | 23.59 | 21.82 | 21.76 | 22.00 | 23.41 | 20.41 | 22.75 |
| $\sigma$ [km] | 3.02 | 2.63 | 3.08 | 2.52 | 2.76 | 2.18 | 1.82 | 2.44 | 3.12 | 2.97 |

**Table 4.** Same as Table 3 but separated for measurements with mountain-wave conditions (wa) and without (nw).

340

The comparison of PSC characteristics for both types of conditions that were derived from our data shows that mountain lee waves, on average produce a very different frequency of occurence of PSC types. Our results focus on the local affect that such waves have on PSCs. Previous studies by others, relying on data from various campaigns, discussed cases that demonstrated

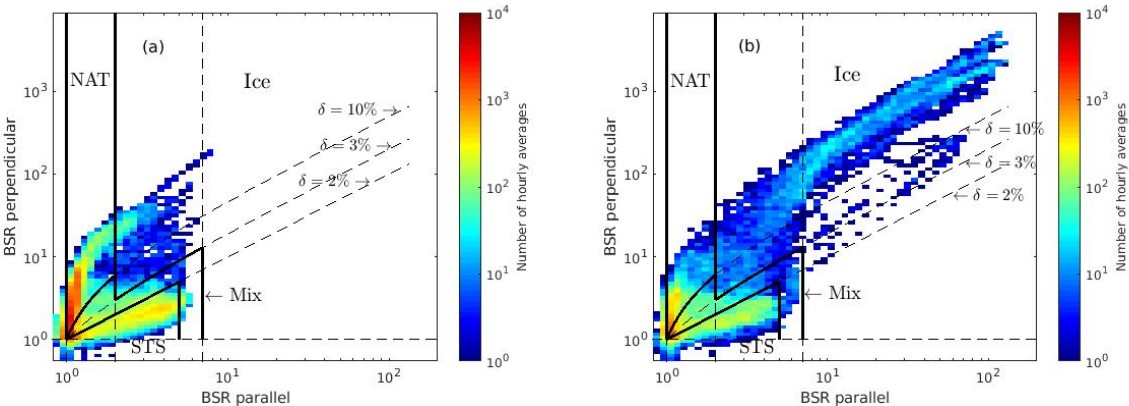

**Figure 8.** Frequency of backscatter ratios of parallel versus perpendicular channel for PSC observations (a) in the absence of mountain lee waves and (b) when waves could propagate to the stratosphere.

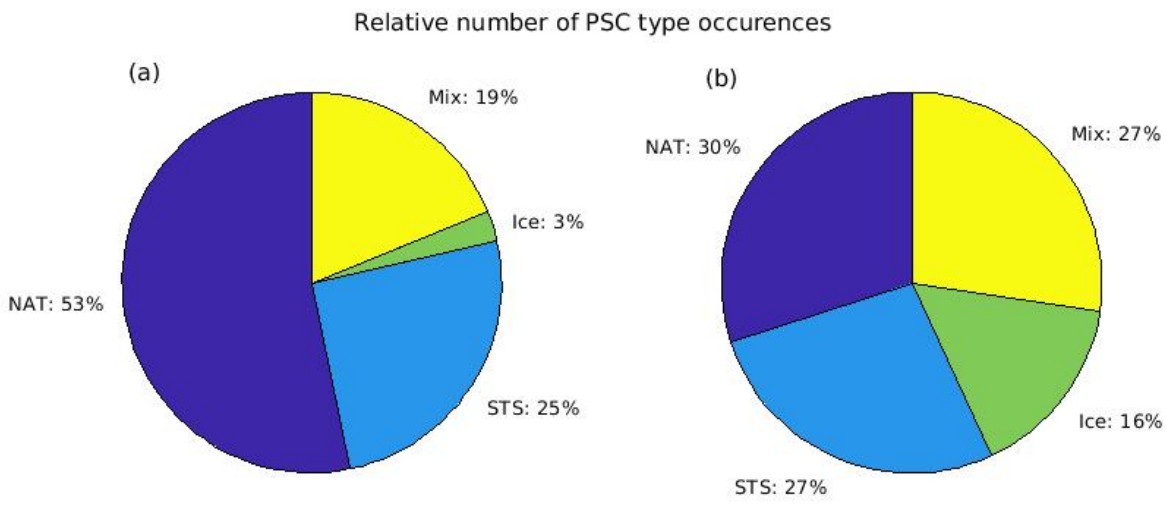

**Figure 9.** Relative distribution of PSC observations (a) in the absence of mountain lee waves and (b) when waves could propagate to the stratosphere (right).

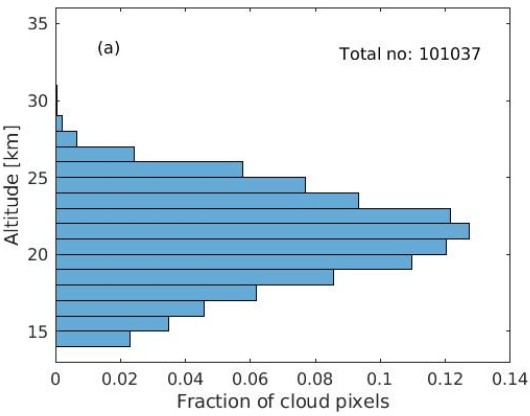 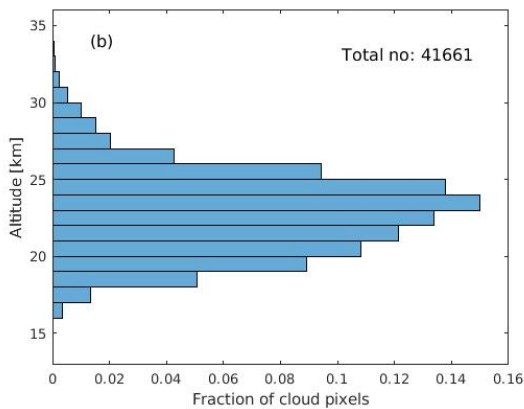

**Figure 10.** Height distribution of PSC observations (a) in the absence of mountain lee waves and (b) when waves could propagate to the stratosphere.

that the effect of mountain lee waves on the chemical and physical characteristics of a PSC is not restricted to the vicinity of the mountain ridge but continues further downstream (e.g. Murphy and Gary, 1995; Riviere et al., 2000; Voigt et al., 2000; Dörnbrack et al., 2002). Ice particles that can form due to adiabatic cooling that comes with the vertical motion due mountain gravity waves were found to act as seeds for NAT particles downstream (Fueglistaler et al., 2003; Eckermann et al., 2009; Alexander et al., 2011). Denitrification that is associated with the formation of NAT particles influences the ozone chemistry (Carslaw et al., 1998b). Hence, a local source of PSCs can affect the stratosphere on a regional scale. An investigation of such effects, however, is beyond the scope of this study.

Data spanning several years or longer can, in principle, allow the identification of a long-term trend if that trend is distinguishable from underlying variablity of the data, i.e. if it has statistical significance. In our data set, however, interannual variations of PSC characteristics are too large to identify any trend.

## 4 Summary

Measurements of PSCs with a backscatter lidar in Kiruna, northern Sweden, were analysed. The data comprises 11 winter seasons, a period sufficiently long to allow for a statistical analysis. Nearly half of all observed clouds consisted of NAT particles while ice clouds were only a small fraction, STS and a mixture of different components making up the rest. The most common altitude for observed PSCs was around 22 km. Those results are in agreement with findings by Pitts et al. (2018) and Blum et al. (2005). When separating lidar observations that were influenced by mountain lee waves and those without such waves present, clearly distinct characteristics become apparent. Ice PSCs were about 5 times as frequent when such waves were present compared to no-wave conditions. On the other hand, NAT PSCs were about half as frequent under wave conditions

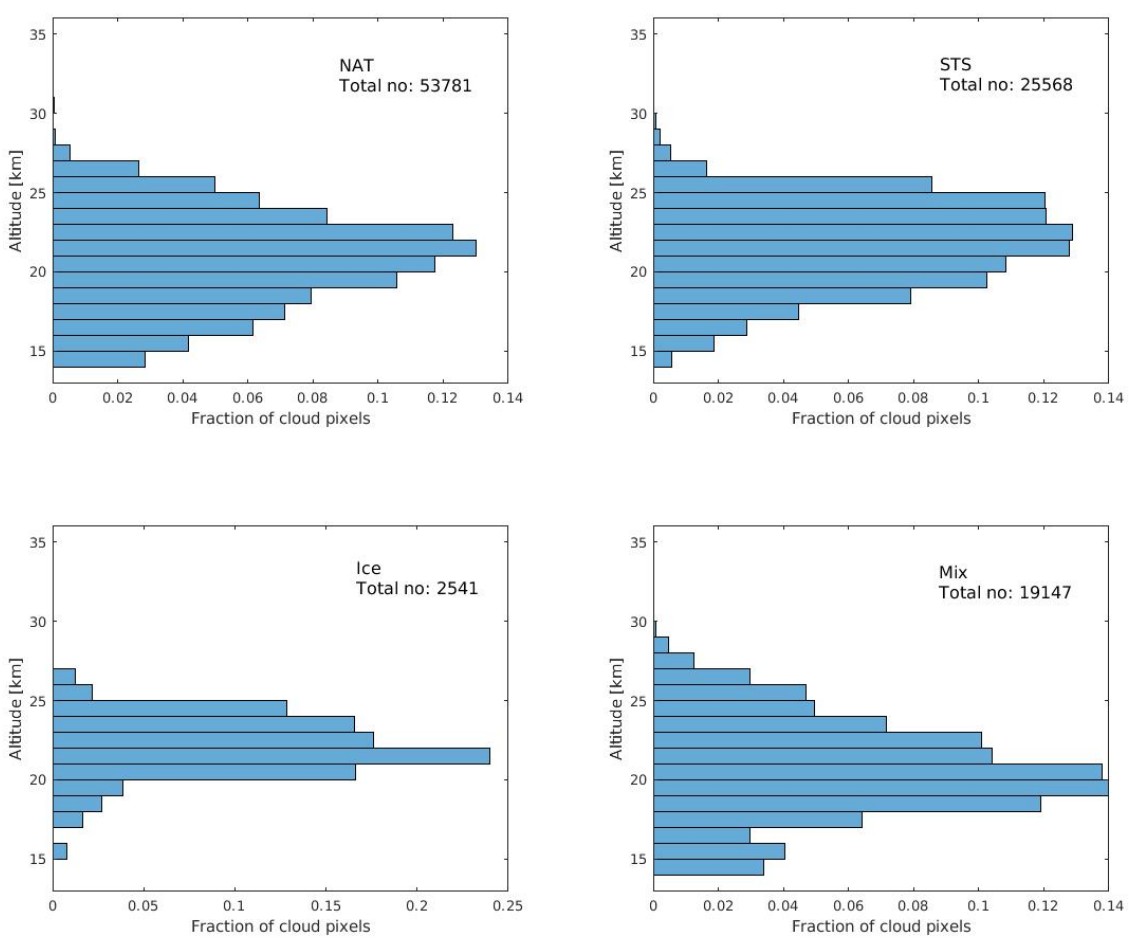

**Figure 11.** Height distribution of detected PSCs for the four types of our classification scheme in the absence of mountain lee waves.

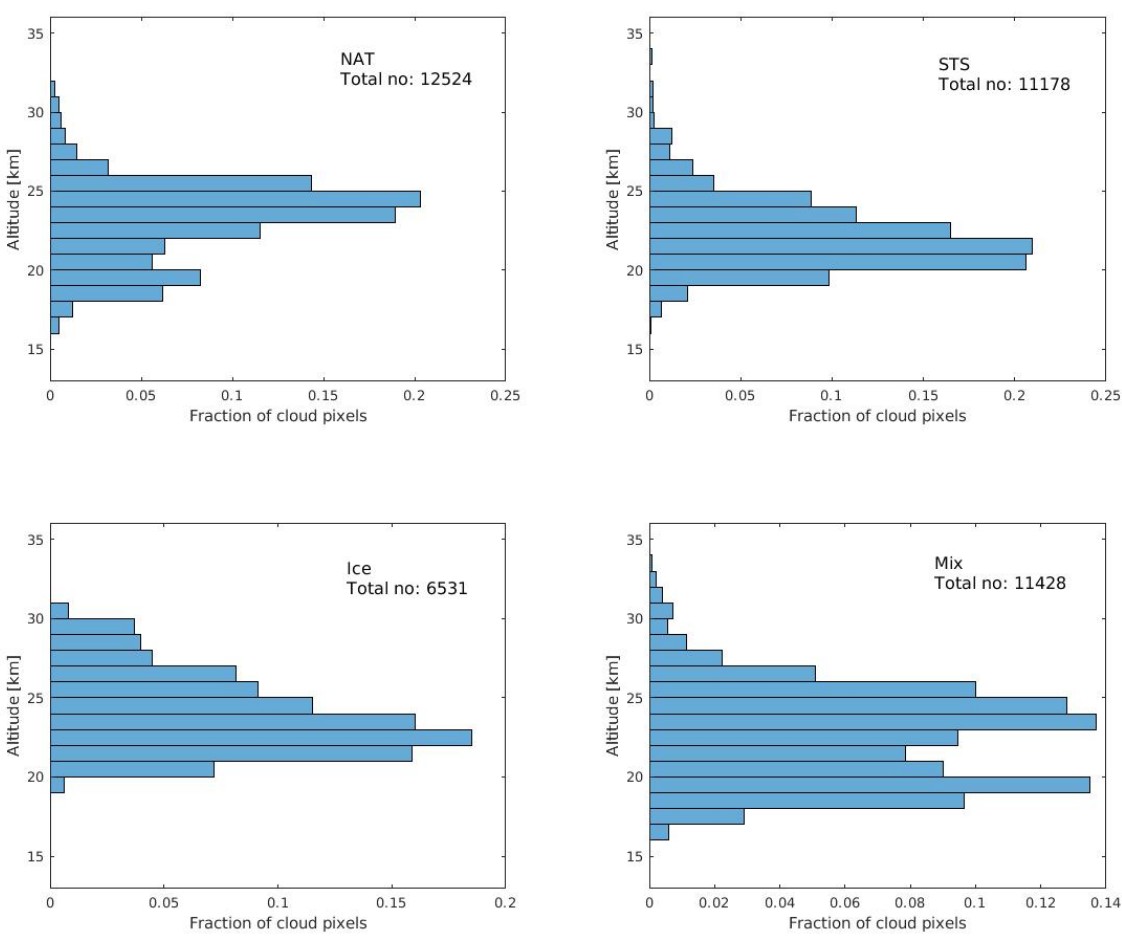

**Figure 12.** Height distribution of detected PSCs for the four types of our classification scheme when mountain lee waves could propagate to the stratosphere.

than without mountain lee waves being present. Observed PSCs were, on average, at 2 km higher altitudes when under the influence of mountain lee waves.

*Data availability.* Lidar data is available from the authors on request. ERA5 datasets can be obtained by approved users from ECMWF (http://www.ecmwf.int).

*Author contributions.* PV and PD performed lidar observations and inverted measurement data. PV did the statistical analysis. PV prepared the manuscript with contributions from PD.

*Competing interests.* The authors declare that they have no conflict of interest.

*Acknowledgements.* We thank the Danish Meteorological Institute for providing PSC and vortex forecasts. We gratefully acknowledge Daria Mikhaylova (IRF) for invaluable help with lidar operations and Oliver Willbrink (Luleå University of Technology, Sweden) for an initial assessment of lidar observation data. We also thank Ronny Engelmann (Leibniz Institute for Tropospheric Research, Germany) for kindly providing instrumentation for testing receiver channel performance. We are grateful to Farah Khosrawi and a second, anonymous reviewer for helpful comments and suggestions.

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
