# Peer review of "Statistical analysis of observations of polar stratospheric clouds with a lidar in Kiruna, northern Sweden"

_EGUsphere, 2022_

## Author Comment (AC1)

**Response to editor for manuscript**

**"Statistical analysis of observations of polar stratospheric clouds
with a lidar in Kiruna, northern Sweden"
P. Voelger and P. Dalin**

The manuscript was modified to include both corrections that the editor
wanted.

---

## Author Comment (AC2)

**Response to reviewers' comments:**

We thank Reviewer #2 for their helpful and constructive comments. In the following we give itemised answers to questions and suggestions by this reviewer.

- Discussion too succinct: The discussion of results was expanded by adding new figures and by discussing the statistics in more detail. We compared our results with other, similar studies. However, we are not aware of many long-term studies of Arctic PSCs. We included the ones we know of. Due to different stratospheric conditions we omitted a comparison with studies of Antarctic PSCs. Some comparison with CALIPSO is included in the study. The major problem we see for performing more detailed comparisons with CALIPSO is that the satellite's swaths are most often at not insignificant distances from our location. When no waves are present a comparison can be feasible (Achtert et al., 2011). With mountain lee waves present observed PSC characteristics can be very different. We contend that we pointed this out in the present study.

- Line 76: It should in principle be possible to combine radar and lidar data for studies of mountain-wave-induced PSCs. However, one has to be aware of a few obstacles. While the radar can detect wave activity up to the UTLS region, it is not set up for measurements in PSC altitudes. It cannot detect clouds directly but observed waves, winds, and turbulence. Hence, the radar can indirectly observe convective clouds, and detect vertical propagation of waves in the troposphere. The stratosphere is too well stratified to yield significant backscatter signals. Therefore, the radar cannot observe PSCs when conditions are bad for lidar measurements. Moreover, radar and lidar are appr. 30 km apart and don't observe the same volume.

- Figure 2 caption: We changed the names of PSC types to 'NAT', 'STS' and 'ice' in the caption and throughout the text to be consistent with current nomenclature.

- Figure 2: The figure was modified to show boundaries of the PSC types. Old Fig. 7 (now Fig. 8) was modified in the same way.

- Figure 2 and Figure 7: Old Fig. 7 is now Fig. 8. We contemplated using the same set of parameters as Tritscher et al. (2021) and others, but ultimately decided against it. The main reason is that doing so would require further steps of data handling which potentially could induce additional errors. Another motivation to keep the current set of parameters was to facilitate a comparison with Blum et al. (2005) whose study is based on measurements in Northern Sweden, as well.

- Figure 4 and 9: Old Fig. 9 is now Fig. 10. We normalised all plots showing height distributions. The plots now also show the total number of cloud pixels that are included in the respective plot. New figures were added (Figs. 5, 11, and 12) that show height distributions for the individual types of our PSC classification. These figures are normalised as well.

- Line 155: L. 156-157 already stated that cloud pixel that are classified as one particular type are not necessarily pure but can contain minor portions of compounds that are characteristic for other types. More explanation has been added which aims specifically at the issue of hard or fuzzy boundaries.

- Line 220 and following: We added some sentences that address mountain-wave seeding of PSCs. This, however, is not the intended subject of this study. Therefore, we want to keep this part relatively short. It is an interesting idea for a future study.

- Figure 6: Now Fig. 7. We added a line that gives the fraction of observations that are favourable for mountain waves. We think that it is still useful to have the absolute numbers, as well, therefore kept them.

- Figure 7: Now Fig. 8. Boundaries for the different PSC types were added.

- Figure 9: Now Fig. 10. New figures were added (Figs. 11 and 12) that show the height distributions of for each PSC type, both with and without mountain lee wave conditions present.

- Figure 9b: In our understanding the lifting of air parcels in the wave will happen adiabatically, hence lead to a cooling of the air parcel. The cooling makes it more likely that formation temperatures for PSCs are reached. Hence, we expect PSCs more likely to form at the wave crest. This explanation was included in the manuscript (l. 300 and following).

45
- Data availability: As far as we understand, the statement 'lidar data is available on request' is consistent with ACP's policies. We are aware that a doi number would be much preferable. However, while our institution wants to get there in the (hopefully) not too far future, this is currently not available, unfortunately. Nevertheless, due to Swedish regulations for governmental institutions we are obligated to provide our lidar data to anyone who is interested in them.

**References**

50  Achtert, P., Khosrawi, F., Blum, U., and Fricke, K. H.: Investigation of polar stratospheric clouds in January 2008 by means of ground-based and spaceborne lidar measurements and microphysical box model simulations, J. Geophys. Res., 116D, D07 201, https://doi.org/10.1029/2010JD014803, 2011.

Blum, U., Fricke, K. H., Müller, K. P., Siebert, J., and Baumgarten, G.: Long-term lidar observations of polar stratospheric clouds at Esrange in northern Sweden, Tellus B, 57, 412–422, https://doi.org/10.3402/tellusb.v57i5.16562, 2005.

55  Tritscher, I., Pitts, M. C., Poole, L. R., Alexander, S. P., Cairo, F., Chipperfield, M. P., Groß, J.-U., Höpfner, M., Lambert, A., Luo, B., Molleker, S., Orr, A., Salawitch, R., Snels, M., Spang, R., Woiwode, W., and Peter, T.: Polar stratospheric clouds: Satellite observations, processes, and role in ozone depletion, Rev. Geophys., 59, 1–81, https://doi.org/10.1029/2020RG000702, 2021.

---

## Author Comment (AC3)

**Response to reviewers' comments:**

We thank Farah Khosrawi for her helpful and constructive comments. In the following we list our answers to specific suggestions and questions by the reviewer.

**Technical corrections:**

We included all suggestions except the last two which addressed the spelling of titles of references. In those cases we kept our original spellings, since they are identical to the spellings that were used in those titles. Yet, we are aware that they may appear uncommon.

**Specific comments:**

– P2, L32: We replaced "clouds" by "PSC".

– P2, L34: The sentence was rewritten and is now: "Over the period of operation the lidar data has been used to generate multiple global PSC climatologies with the latest version covering 11 years of observations until 2017 (Pitts et al., 2018)."

– P2, L44: The study by Tencé et al. (2022) is now mentioned in this section.

– P2, L54ff: A sentence was added L50ff with the purpose to explain the different scales. However, it has to be emphasized that 'hard' boundaries between types rarely exist.

– P3, L66: In this sentence the word "cloud" was replaced by "PSC".

– P3, L68: The sentence was rewritten: "Chemical reactions on the surfaces of PSC particles can then cause large-scale ozone depletion."

– P3, L78: We included three more recent references, one of them a publication in connection with the RECONCILE campaign.

– P4, L. 95ff: The numbers of measurement days and hours are discussed in the beginning of section 3. Here, we added a sentence to specifically point this out.

– P5, Sect. 3: Examples for warm and cold winters have been added. Winters 2014/15 (warm) and 2015/16 (cold) can be seen as examples how PSCs are more frequently present during a cold winter. Additionally, winters 2012/13 (warm) and 2013/14 (cold) are mentioned to show that the number of measurement opportunities for a ground-based lidar are strongly influenced by tropospheric conditions.

– P5, L135: We added some more explanation. Whether stratospheric conditions are favourable for the existence of PSCs depends on the prevailing temperature. During winter months the Danish Meteorological Institute compiles analyses and forecasts for the Arctic stratosphere based on ECMWF data. Those data are now compared with our measurement statistics. We are not aware of publications that specifically discuss the seasonal distribution of PSCs over Kiruna.

– P6, L149: We extended this part to include more reasoning why our classification scheme is similar to the one by Blum et al. (2005).

– P6, L164: It was not our intention to claim that $T_{NAT} < T_{STS}$. Relative to $T_{ice}$ it is commonly assumed that $T_{NAT}$ is appr. $7K$ higher (Hanson and Mauersberger, 1988), while $T_{STS}$ is appr $4K$ higher (Carslaw et al., 1995). The statement was rewritten to avoid misunderstandings.

- P7, Figure 2 caption: NAT, STS and ice were added in parentheses.

- P8, Figure 4: We included an additional figure that shows plots with height distributions for each of the PSC types of our classification scheme. The new figure is Fig. 5.

- P9, Figure 5: This is Fig. 6 now. The legend was modified to include NAT, STS, and ice.

- P9, L198: It is very often implicitly assumed that reanalysis datasets like ERA5 provide a reasonably good description of the atmospheric state at a certain time and location. Comparisons with radiosondes (see e.g. Graham et al. (2019) and references therein) and, in the case of winds, with radar data (e.g. Sivan et al., 2021) appear to confirm this. Concerning our location, an additional investigation how ERA5 compares with measurements by radiosondes that sporadically are launched from Esrange (appr. 30 km from the location of our lidar) or with winds derived from the wind profiling radar at Esrange are beyond the scope of this study. We assume that the good representation of atmospheric conditions at other locations that ERA5 provides also is true for Kiruna.

- P11, L226: We believe that the larger heights of wave PSCs are due to adiabatic cooling as a result of lifting of air parcels in the wave motion. A similar phenomenon can be observed in tropospheric clouds in the lee of mountains.

- P11, L229 and L231: The part was rewritten. The purpose of this paragraph was (and is) to stress that we cannot detect in our data a trend over time, the reason being the high interannual variability of the Arctic polar vortex. A review of the current status of research on the impact of climate change on the stratosphere is not the aim of this study.

- P12, Figure 9: This is now Fig. 10. New figures were added, Figs. 11 and 12, that show the height distributions for the individual types of our PSC classification.

**References**

Blum, U., Fricke, K. H., Müller, K. P., Siebert, J., and Baumgarten, G.: Long-term lidar observations of polar stratospheric clouds at Esrange in northern Sweden, Tellus B, 57, 412–422, https://doi.org/10.3402/tellusb.v57i5.16562, 2005.

Carslaw, K. S., Clegg, S. L., and Brimblecombe, P.: A Thermodynamic Model of the System HCl-HNO3-H2SO4-H2O, Including Solubilities of HBr, from <200 to 328 K, J. Phys. Chem., 99, 11 557–11 574, https://doi.org/10.1021/j100029a039, 1995.

Graham, R. M., Hudson, S. R., and Maturilli, M.: Improved performance of ERA5 in Arctic gateway relative to four global atmospheric reanalyses, Geophys. Res. Lett., 46, 6138–6147, https://doi.org/10.1029/2019GL082781, 2019.

Hanson, D. and Mauersberger, K.: Laboratory studies of the nitric acid trihydrate: Implications for the south polar stratosphere, Geophys. Res. Lett., 15, 855–858, https://doi.org/10.1029/GL015i008p00855, 1988.

Pitts, M. C., Poole, L. R., and Gonzalez, R.: Polar stratospheric cloud climatology based on CALIPSO spaceborne lidar measurements from 2006 to 2017, Atm. Chem. Phys., 18, 10 881–10 913, https://doi.org/10.5194/acp-18-10881-2018, 2018.

Sivan, C., Rakesh, V., Abhilash, S., and Mohanakumar, K.: Evaluation of global reanalysis winds and high-resolution regional model outputs with the 205 MHz stratosphere-troposphere wind profiler radar observations, Quart. J. Roy. Meteorol. Soc., 147, 2562–2579, https://doi.org/10.1002/qj.4041, 2021.

Tencé, F., Jumelet, J., Bouillon, M., Cugnet, D., Bekki, S., Safieddine, S., Keckhut, P., and Sarkissian, A.: 14 years of lidar measurements of Polar Stratospheric Clouds at the French Antarctic Station Dumont d'Urville, ACPD, 2022, 1–27, https://doi.org/10.5194/acp-2022-401, 2022.

---

## Author Response (AR3)

**Response to reviewers' comments:**

**Reviewer #1:**

We thank Farah Khosrawi for her additional comments. In the following we list our answers to specific suggestions and comments by the reviewer.

5

**Authors' answers to comments was not uploaded:**
Apologies! We obviously messed up here. For what it's worth, the response is uploaded now and should be possible to find on the discussion page.

**Specific comments:**

10
- P1, L4: We changed the sentence. It now reads: 'The statistical analysis demonstrates that nearly half of all observed PSCs consisted of nitric acid trihydrate (NAT) particles while ice clouds accounted only for a small fraction and the remainder consisted of supercooled ternary solution (STS) and mixtures of different compositions.'.

- P1, L14: We changed the sentence as suggested by the reviewer.

- P1, L15: We replaced 'in' with 'by'.

15
- P4, L113: We deleted 'is'.

- P5, L158: We understand that this sentence would benefit from an extended discussion of the value and the shortcomings of a visual PSC statistic. However, this is beyond the aim of this study, which intends to focus on lidar measurements. We therefore removed the sentence.

- P7, L192: 'as' was added.

20
- P11, Figure 4 caption: 'in' was replaced with 'on'.

- P13, L284: 'Arctic' was corrected.

- P14, L303: Sentence was corrected and now reads '... by large numbers of ...'.

- P19, L336: 'affect' was replaced with 'effect'.

- P19, L341: 'to' was added.

25
- P19, L352: The sentence was changed and reads now: '... while ice clouds accounted only for a small fraction and the remainder consisted of STS and mixtures of different compositions.'.

**Reviewer #2:**

We thank reviewer #2 for their additional comments. In the following we list our answers to specific suggestions by the reviewer (line numbers from the Tracked Changes version, as listed by the reviewer).

30

- Line 202: We added 'as'.

- Line 221: The boundaries of our PSC classification are based on Blum et al. (2005) and similar to theirs but were adapted for the specifications of our lidar. This clarification has been included both in the text and in the capotion for Figure 2.

- Line 242: We extended the sentence based on the suggestions of the reviewer. It reads now: '... from a single location that frequently is situated downwind of a major orographic obstacle, the Scandinavian Mountains.'

- Line 245: Changed to frequently

- Line 278: We extended this part in order to, hopefully, make it appear less hand-wavy. We added sentences that stressed (a) that only a small portion of all observed PSCs were found at these altitudes since temperatures are most often too warm for PSCs tro exist, (b) that, when PSCs can exist, temperatures are often sufficiently low for Ice PSCs to exist, and (c) that this could be an effect of mountain lee waves.

- Line 300: Was corrected.

- Line 332: The sentence reads now: 'Ice clouds accounted for about 1/6 ...'.

- Line 343: The sentence was corrected.

**References**

45   Blum, U., Fricke, K. H., Müller, K. P., Siebert, J., and Baumgarten, G.: Long-term lidar observations of polar stratospheric clouds at Esrange
     in northern Sweden, Tellus B, 57, 412–422, https://doi.org/10.3402/tellusb.v57i5.16562, 2005.

---

## Author Response (AR4)

**Response to editor for manuscript**

**"Statistical analysis of observations of polar stratospheric clouds
with a lidar in Kiruna, northern Sweden"**
**P. Voelger and P. Dalin**

The manuscript was modified to include the missing period in the caption
for Figure 2 that the editor pointed out.